

# Divergent relationship of circulating CTRP3 levels between obesity and gender: a cross-sectional study

Roy Marshal Wagner[1], Kamesh Sivagnanam[2], William Andrew Clark[1] and Jonathan M. Peterson[3,4]

[1] Allied Health Sciences, College of Clinical and Rehabilitative Health Sciences, East Tennessee State University, Johnson City, TN, United States
[2] Department of Internal Medicine, Quillen College of Medicine, East Tennessee State University, Johnson City, TN, United States
[3] Department of Biomedical Sciences, Quillen College of Medicine, East Tennessee State University, Johnson City, TN, United States
[4] Department of Health Sciences, East Tennessee State University, Johnson City, TN, USA

Corresponding author
Jonathan M. Peterson,
petersonjm1@etsu.edu

## ABSTRACT

C1q TNF Related Protein 3 (CTRP3) is a novel adipose tissue derived secreted factor, or adipokine, which has been linked to a number of beneficial biological effects on metabolism, inflammation, and survival signaling in a variety of tissues. However, very little is known about CTRP3 in regards to human health. The purpose of this project was to examine circulating CTRP3 levels in a clinical population, patients with symptoms requiring heart catheterization in order to identify the presence of obstructive coronary artery disease (CAD). It was hypothesized that serum CTRP3 levels would be decreased in the presence of CAD.

**Methods.** Body mass index (BMI), diabetes status, and plasma samples were collected from 100 patients who were >30 years of age and presented at the East Tennessee State University Heart Clinic with symptoms requiring heart catheterization in order to identify the presence of cardiovascular blockages ($n = 52$ male, $n = 48$ female). Circulating CTRP3 levels were quantified using commercially available ELISA.

**Results.** Circulating CTRP3 levels had no relationship to the presence of CAD regardless of gender. However, circulating concentrations of CTRP3 were significantly higher in normal weight (BMI < 30) females ($0.88 \pm 0.12$ µg/ml) compared with males ($0.54 \pm 0.06$ µg/ml). Further, obesity (BMI > 30) resulted in an increase in circulating CTRP3 levels in male subjects ($0.74 \pm 0.08$ µg/ml) but showed a significant decrease in female subjects ($0.58 \pm 0.07$ µg/ml). Additionally, there was a significant reduction in circulating CTRP3 levels in female subjects who were diagnosed with Type 2 diabetes compared with patients without ($0.79 \pm 0.08$ vs. $0.42 \pm 0.10$ µg/ml). There was no relationship between diabetes status and circulating CTRP3 levels in male subjects.

**Conclusion.** Circulating CTRP3 levels had a different relationship with diabetes and obesity status between male and female patients. It is possible that circulating CTRP3 levels are controlled by hormonal status, however more research is needed to explore this relationship. Nevertheless, future studies examining the relationship between CTRP3 levels and disease status should treat gender as an independent variable.

## INTRODUCTION

Cardiovascular disease is the leading cause of death worldwide (*Gaziano et al., 2006*). Recent studies have shown that obesity and diabetes mellitus are two of the largest risk factors for the development of coronary heart disease, even more so than cigarette smoking, dyslipidemia, and even hypertension (*Smith Jr, 2007*). Obesity is a chronic disease affecting over one-third of US adults (*Ben-Menachem, 2007*; *Flegal et al., 2012*; *Ogden et al., 2012*). However, the mechanisms by which obesity and diabetes mellitus increase the prevalence of cardiovascular disease are still being actively investigated.

Dysfunctional adipose tissue represents a novel paradigm that mechanistically links obesity to a variety of diseases such as cardiovascular disease. Traditionally adipose tissue is thought of as solely a storage vessel for excess lipids; however, adipose tissue secretes a number of factors, which have essential endocrine roles in regulating biological functions (*Garg, 2006*; *Jankovic et al., 1969*; *Wang, Jin & Wang, 1986*). These adipose tissue-derived secreted factors are collectively called adipokines. Efforts to identify novel adipokines have led to the discovery of a family of secreted proteins, designated as C1q TNF-related protein 1-15 (CTRP1-15) (*Peterson, Wei & Wong, 2010*; *Seldin et al., 2012*; *Seldin, Tan & Wong, 2014*; *Wei et al., 2012*; *Wei, Peterson & Wong, 2011*; *Wei et al., 2013*; *Wong et al., 2009*; *Wong et al., 2008*). The CTRP proteins, adiponectin, TNF-alpha, as well as other proteins with the C1q domain together are grouped together as the C1q/TNF superfamily (*Wong et al., 2004*). Of these proteins CTRP3 (synonyms CORS-26, cartducin and cartonectin) has been demonstrated to protect against ischemia-induced cardiac damage (*Wu et al., 2015*; *Yi et al., 2012*) and circulating CTRP3 levels were recently found to be reduced in patients with acute coronary syndrome (*Choi et al., 2014*).

Previous research has shown that circulating CTRP3 levels are reduced with conditions associated with cardiovascular events such as obesity and metabolic syndrome (*Ban et al., 2014*; *Deng et al., 2015*; *Qu et al., 2015*; *Wolf et al., 2015*). Conversely, CTRP3 levels have also be reported to be elevated with obesity and/or metabolic syndrome (*Choi et al., 2012*). Regardless, the potential use of circulating CTRP3 levels as a biomarker for risk or the presence of cardiovascular disease has not been established. These data led to our initial hypothesis: patients with obstructive coronary artery disease (CAD) will have reduced circulating CTRP3 levels compared with patients without CAD. Although our data did not support our initial hypothesis, we did find a number of novel observations regarding the relationship of circulating CTRP3 levels, gender and obesity that sheds light on some of the contradictory reports of the CTRP3 in the literature.

## METHODS

### Ethics statement

All study subjects provided written informed consent for the study procedures. The study was approved by the Institutional Review Board of East Tennessee State University's Office of Research and Sponsored Programs (IRB #0313-35s, 2013, ETSU/VA Medical IRB Board).

**Table 1** Categorical population overview.

|  | Overall | Male | Female |
|---|---|---|---|
| Total | 100 | 52 | 48 |
| Obese (BMI > 30) | 57 | 24 | 33 |
| Type 2 Diabetes | 30 | 15 | 15 |
| Smoker | 53 | 30 | 20 |
| Presence of CAD | 29 | 18 | 10 |

## Study design and participants

This was a cross-sectional study conducted from June 2013 to June 2015 at the East Tennessee State University Heart Clinic, an affiliate of James H Quillen College of Medicine Division of Cardiology. Inclusion criteria included: patients with symptoms requiring heart catheterization in order to identify the presence of obstructive coronary artery disease (CAD). Exclusion criteria included known history of CAD, Coronary artery bypass grafting (CABG), coronary stent placement, type 1 diabetes and/or <30 years of age.

## Clinical and laboratory measurements

Self-reported measures of age, sex, ethnicity, and smoking status were obtained from each participant. Clinical data (diagnosis of Type 2 Diabetes, Blood pressure, height, and weight) were obtained as part of routine medical examination. Body mass index (BMI) was calculated as weight/height$^2$ (kg/m$^2$). Blood samples for lipid analysis were obtained through a commercial lab as part of the routine medical visit. For plasma collection samples were collected from non-fasting trial participants into 5-mL EDTA vacutainers, immediately refrigerated, transported to the laboratory, and processed within 24 hours after withdrawal. Samples were centrifuged at room temperature for 10 min at 3,000 × g, the plasma (supernatant) was decanted using a Pasteur pipette and transferred to a 3 mL glass amber vial and stored at −30 °C until further analysis. Human CTRP3 levels were measured by ELISA (Aviscera Bioscience Cat#SK00082) according to manufacturer's protocol, with an intra-assay coefficient of variation of less than 10%. Samples with CTRP3 levels below the detection limit of the assay were assigned the lowest detectable value (0.077 μg/ml). Glucose levels were measured by commercial glucose analyzer (One touch Ultra).

## Statistical analysis

Frequencies, means, and standard deviations were calculated for descriptive analyses (Table 1). Our analysis considered gender as an independent variable and a 2-way ANOVA was performed for categorical variables (presence of blockage, diabetes, obesity (BMI > 30), smoker) with Fisher's LSD test analysis for multiple comparisons. Spearman's correlations were perform for continuous variables. Significance was set at $p \leq 0.05$ and a statistical trend was defined as $0.05 < p \leq 0.09$. Statistical analyses were conducted using GraphPad 6.0.

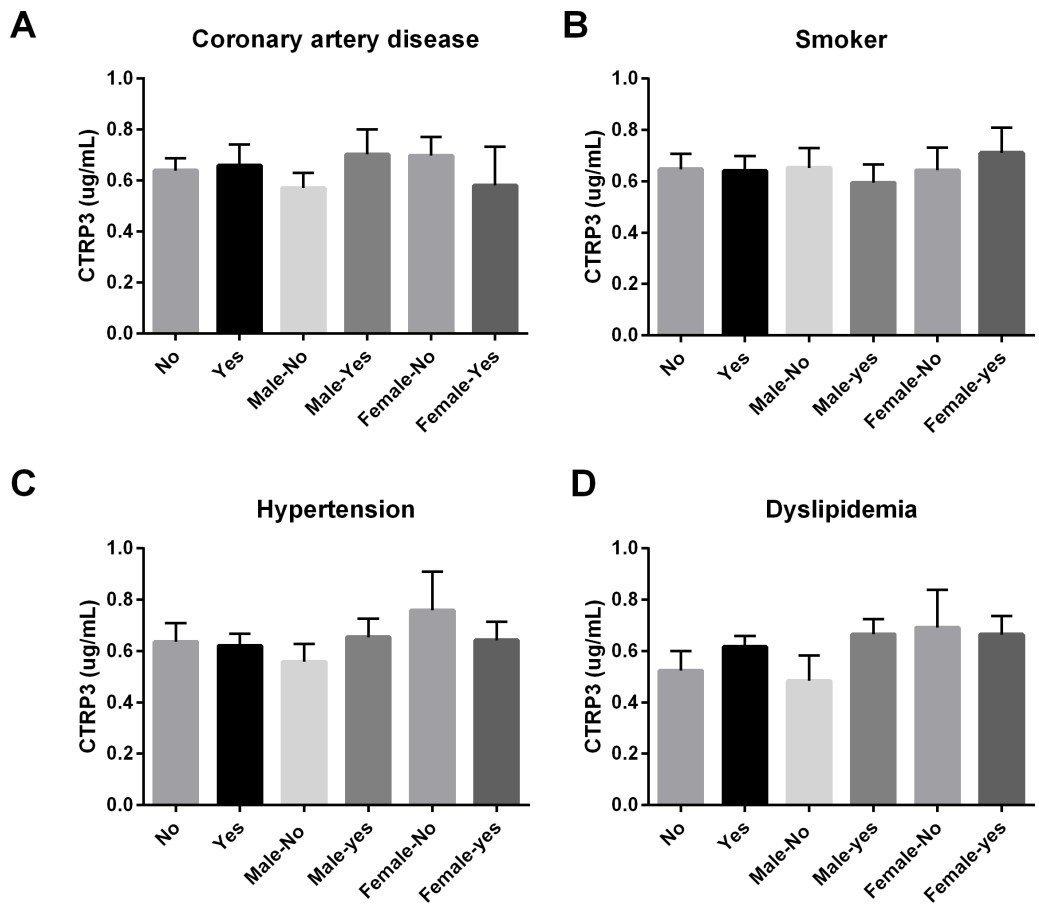

**Figure 1** Circulating CTRP3 levels were compared across the population and within gender based on the: (A) presence of coronary artery disease (CAD); (B) tobacco use; (C) hypertension; or (D) dyslipidemia. No differences in circulating CTRP3 levels were observed within this population. Data are presented as mean ± SEM. Significance was set at $p < 0.05$.

## RESULTS AND DISCUSSION

### Study population

There were a total of 100 participants ($n = 52$ male, $n = 48$ female), 28 were positive for CAD. The remaining 72 participants did not have significant findings during catheterization. The average age of the study subjects was $57.5 \pm 12.6$ years, and their average BMI was $33.09 \pm 7.9$ kg/m$^2$. Additional descriptive statistics are listed in Table 1.

### Relationship between CTRP3 and categorical variables

In this cross-sectional study of patients with a variety of symptoms, we initially hypothesized that because CTRP3 has cardioprotective properties lower circulating CTRP3 levels would be associated with CAD. Associations between circulating CTRP3 levels and anthropomorphic measurements, presence of diabetes, gender, and obesity were also investigated. No differences were observed in this population regarding circulating CTRP3 levels and CAD, dyslipidemia, hypertension, or tobacco use (Fig. 1). There was not a significant difference in circulating CTRP3 levels between male and female subjects (Fig. 2A),

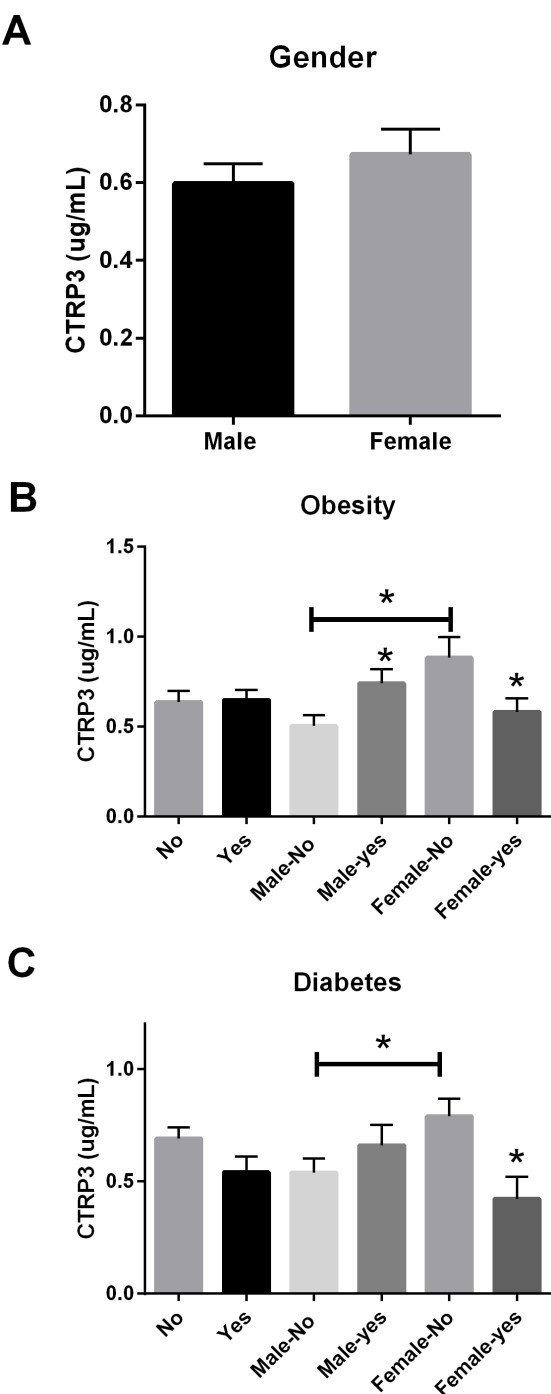

**Figure 2  Circulating CTRP3 level by gender, obesity, and type 2 Diabetes.** (A) Circulating CTRP3 levels were similar between all male and female subjects. (B) CTRP3 levels were different when examined in the presence of obesity. Specifically, CTRP3 levels were higher in lean females compared with lean male subjects, but CTRP3 levels were reduced in the obese females. On the other hand, CTRP3 levels were significantly higher in obese male subjects. (C) The diagnosis of Type 2 diabetes had no association with circulating CTRP3 levels in male subjects but diabetic females had ∼50% lower level of circulating CTRP3 than non-diabetic females. Data are presented as mean ± SEM. Significance was set at $p < 0.05$.

**Table 2  Overall relationship to circulating CTRP3 levels.**

|  | P-value | r-value | Mean |  | SD |
|---|---|---|---|---|---|
| AGE (yrs) | 0.089 | −0.137 | 57.46 | ± | 12.66 |
| Total Cholesterol (mg/dL) | 0.123 | −0.131 | 171.60 | ± | 43.19 |
| HDL (mg/dL) | 0.111 | −0.137 | 39.35 | ± | 9.66 |
| LDL (mg/dL) | 0.069 | −0.169 | 100.40 | ± | 42.00 |
| TRIGLY (mg/dL) | 0.430 | −0.020 | 163.50 | ± | 88.01 |
| Glucose (mg/dL) | 0.178 | 0.094 | 90.65 | ± | 53.38 |
| BMI | 0.276 | 0.061 | 33.09 | ± | 7.89 |

however CTRP3 levels were significantly elevated in lean females compared with lean male subjects (Fig. 2B). Further, circulating CTRP3 levels were significantly higher in obese compared with lean males and conversely, CTRP3 levels were significantly lower in obese females compared with lean females (Fig. 2B). While the presence of diabetes had no association with circulating CTRP3 levels in male subjects, diabetic females had close to a 50% lower level of circulating CTRP3 (Fig. 2C). Although this study was limited to this specific clinical subgroup (those with symptoms requiring catheterization), there were a number of interesting and novel observations made that shed light on the conflicting findings of the relationship between diabetes and obesity have with circulating CTRP3 levels (Ban et al., 2014; Choi et al., 2012; Deng et al., 2015; Qu et al., 2015; Wolf et al., 2015).

## Relationship between CTRP3 and continuous variables

We examined the relationship between circulating CTRP3 levels and age, cholesterol, triglycerides, glucose, and BMI across the population and did not observe significant relationships (Table 2). We then repeated the analysis independently with each gender and observed that in the males there was a significant positive correlation between BMI and circulating CTRP3 levels in males (Table 3), whereas no significant correlations were observed in females (Table 4).

Our initial analysis found few overall correlations between circulating CTRP3 levels and any of the variables listed. Although disappointing a thorough examination of the literature highlighted that circulating CTRP3 levels are higher in females than males (Choi et al., 2012; Peterson, Wei & Wong, 2010; Wolf et al., 2015). This observation combined with new NIH guidelines for considering sex as a biological variable caused us to reexamine our data using gender as an independent variable. After this analysis we observed a number of significant differences between the gender regarding circulating CTRP3 levels and obesity, dislipidemia, and Type 2 diabetes. To the best of our knowledge this is the first study to independently examine CTRP3 levels by both gender and obesity. Although it has been previously reported that circulating CTRP3 levels are higher in females than males, our finding of a divergent association of between the genders and obesity, diabetes, and dislipidemia are completely novel observations. Furthermore, these observations shed light on some of the discrepancy in the literature. For example, Wolf et al. (2015) measured circulating CTRP3 levels in lean and obese subjects and showed CTRP3 levels were reduced with obesity. However, 74% of the study population and 89% of the obese subjects were

**Table 3  Male relationship to circulating CTRP3 levels.**

|  | P-value | r-value | Mean | | SD |
|---|---|---|---|---|---|
| AGE (yrs) | 0.281 | −0.084 | 56.06 | ± | 13.6 |
| Total Cholesterol (mg/dL) | 0.166 | −0.157 | 160.10 | ± | 39.6 |
| HDL (mg/dL) | 0.134 | −0.180 | 36.00 | ± | 8.3 |
| LDL (mg/dL) | 0.168 | −0.158 | 89.21 | ± | 39.6 |
| TRIGLY (mg/dL) | 0.351 | −0.063 | 169.1 | ± | 86.4 |
| Glucose (mg/dL) | 0.354 | 0.054 | 90.00 | ± | 54.2 |
| BMI | 0.020 * | 0.291 | 31.73 | ± | 7.5 |

**Notes.**

*indicates significance and was set at $p < 0.05$.

**Table 4  Female relationship to circulating CTRP3 levels.**

|  | P-value | r-value | Mean | | SD |
|---|---|---|---|---|---|
| AGE (yrs) | 0.130 | −0.165 | 58.42 | ± | 11.40 |
| Total Cholesterol (mg/dL) | 0.193 | −0.141 | 184.40 | ± | 43.39 |
| HDL (mg/dL) | 0.193 | −0.141 | 42.85 | ± | 9.82 |
| LDL (mg/dL) | 0.289 | −0.092 | 112.30 | ± | 41.93 |
| TRIGLY (mg/dL) | 0.263 | 0.103 | 160.40 | ± | 90.18 |
| Glucose (mg/dL) | 0.252 | 0.099 | 91.13 | ± | 53.31 |
| BMI | 0.215 | −0.116 | 34.42 | ± | 8.22 |

female, therefore our findings are consistent in that obese females have lowers circulating CTRP3 levels. *Choi et al. (2014)* reported no significant association between CTRP3 levels and BMI in a study with a higher proportion of males (244/118 male/female) but did report negative association with CTRP3 and BMI in another study of 349 subjects (134/215 male/female) (*Choi et al., 2012*). However, in both manuscripts the authors analyzed the association between CTRP3 and BMI across the population and the opposing trajectories of gender as a factor potentially masked the discovery of significant associations. Direct query to the authors for clarification, confirmed that significant differences in CTRP3 levels were observed between   the genders (K Choi, pers. comm., 2016).

We found that in female subjects the presence of diabetes was associated with lower circulating CTRP3 levels. Interestingly, this observation is supported by the work of *Tan et al. (2013)* who reported that women with polycystic ovary syndrome had lower levels of CTRP3 than control subjects. Polycystic ovary syndrome is an endocrine system disorder associated with obesity, diabetes, dyslipidemia, and cardiovascular complications. Further, in this population CTRP3 levels were restored with Metformin treatment, which is the most common drug used to treat type 2 diabetes (*Tan et al., 2013*). On the other hand *Choi et al. (2012)* examined circulating CTRP3 levels in partially female dominated subject population (349 subjects; 134/215 male/female) and reported that CTRP3 levels were increased in patients with type 2 diabetes. This finding is contradictory to our findings and highlights that there is still much unknown regarding the regulation of circulating CTRP3 levels. Further, although authors reported that women had a ∼25% higher level of

CTRP3 than men (*Choi et al., 2012*) they did not analyze the relationship between obesity and CTRP3 levels independently by gender.

CTRP3 is thought to improve insulin sensitivity and metabolic function, particularly in the liver (*Peterson, Wei & Wong, 2010*). Additionally, CTRP3 has been shown to reduce myocardial damage following ischemia (*Wu et al., 2015*; *Yi et al., 2012*). Currently there are no clinically defined biological cutoffs for circulating CTRP3 levels, however, our work indicates that future studies regarding the clinical significance of CTRP3 must evaluate difference between the genders separately as the effect of metabolic dysfunction (obesity and/or diabetes) has conflicting effect on circulating CTRP3 levels between the genders.

### Study limitations and conclusion

One limitation of this study is that it comprised of only a clinical cardiovascular population recruited from a single region and our observations may not apply to the general population. The second and primary limitation of this study is that we were only able to obtain non-fasting samples from the study population, which significantly increased the variability of the data and our results may not have been observed in a fasting population. This is especially important to note considering that, unlike many adipokines, CTRP3 levels have been reported to increase with fasting (*Peterson, Wei & Wong, 2010*). Further, the type and sensitivity of the antibodies used in the ELISA, may also contribute to variability between studies. Nevertheless, our results present a novel finding regarding circulating CTRP3 levels and raises interesting questions on the factors that regulate CTRP3 concentrations in human subjects. Further research is needed to clarify how CTRP3 is regulated and previous studies performed with data from both genders combined should be reexamined with gender treated as an independent variable. Future studies need to be performed to elucidate the mechanism by which the novel adipokine CTRP3 is regulated and how gender affects this regulation. Our findings support the newly released National Institutes of Health (NIH) reproducibility guidelines the "NIH expects that sex as a biological variable (SABV) will be considered in research designs, analyses..." (*NIH, 2016*).

### Funding

This research was supported in part National Institute On Alcohol Abuse And Alcoholism of the National Institutes of Health under Award Number R03AA023612, by the National Institutes of Health Award Number C06RR0306551, East Tennessee State University Research Development Committee (E82262), and College of Clinical & Rehabilitative Health Sciences, ETSU, Dean's Student Research Grant. The funders had no role in study design, data collection and analysis, decision to publish, or preparation of the manuscript.

### Grant Disclosures

The following grant information was disclosed by the authors:
National Institute On Alcohol Abuse And Alcoholism of the National Institutes of Health: R03AA023612.

National Institutes of Health: C06RR0306551.
East Tennessee State University Research Development Committee: E82262.
College of Clinical & Rehabilitative Health Sciences, ETSU, Dean's Student.

## Competing Interests

The authors declare there are no competing interests.

## Author Contributions

- Roy Marshal Wagner performed the experiments, analyzed the data, wrote the paper, reviewed drafts of the paper.
- Kamesh Sivagnanam performed the experiments, contributed reagents/materials/analysis tools, reviewed drafts of the paper.
- William Andrew Clark conceived and designed the experiments, analyzed the data, reviewed drafts of the paper.
- Jonathan M. Peterson conceived and designed the experiments, analyzed the data, wrote the paper, prepared figures and/or tables, reviewed drafts of the paper.

## Human Ethics

The following information was supplied relating to ethical approvals (i.e., approving body and any reference numbers):

All study subjects provided written informed consent for the study procedures. The study was approved by the Institutional Review Board of East Tennessee State University's Office of Research and Sponsored Programs (IRB #0313-35s, 2013, ETSU/VA Medical IRB Board).

## Data Availability

The raw data has been supplied as a Data S1.

## Supplemental Information

Supplemental information for this article can be found online at http://dx.doi.org/10.7717/peerj.2573#supplemental-information.

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
