# Peer review of "Divergent relationship of circulating CTRP3 levels between obesity and gender: a cross-sectional study"

_PeerJ, doi:10.7717/peerj.2573_

## Round 0.1 · original submission · Minor Revisions

Please consider suggestions from reviewer 3. This will further improve the already good quality of the manuscript.

Reviewer 1 ·

Basic reporting

The overall conclusion of the paper is supported by the data presented. The authors clearly articulated the limitation of the study.

Experimental design

No issues with experimental design. The authors use the appropriate statistics to analyze their human data.

Validity of the findings

The findings are valid and limitations of the study are clearly indicated

Additional comments

This human study adds to the growing literature on CTRP3, a secreted plasma protein produced by adipose and other tissues. While the sample size is small, the data indicates interesting sex-dependent differences in circulating levels of CTRP3 between male and female under different (patho)physiological states.

Reviewer 2 ·

Basic reporting

No Comments

Experimental design

No Comments

Validity of the findings

No Comments

Reviewer 3 ·

Basic reporting

Well written, good language. The introduction and background provide context for study. Figures and tables well labeled.

Experimental design

Although stated, the serum specimens are not-fasting, and this is different methodology than used in all other studies examining circulating CTRP3 in humans, making these results very difficult to compare to other authors' findings.

Validity of the findings

1) Although the author's found discordant results with previous studies examining CTRP3 in CAD, obesity, diabetes and gender, interpretation has to be cautious because these serum samples were not fasting (different than previous studies), and the author's did not mention which CTRP3 Elisa was used in each study, which can also affect comparison between study results.
2) The author's state that the average BMI is 24.9kg/m22, but with 57% of the patients obese with BMI >30kg/m2, can the authors clarify the breakdown of BMI in this study cohort. ( also in Table 2, the mean BMI listed as 33.09kg/m2).

Additional comments

Although the authors argue that their findings are novel, a lot of similar findings are embedded in prior publications.
Overall, well written manuscript. Add into discussion that results can be different based on ELISA used and because samples not fasting.

---

## Round 0.2 · accepted · Accept

The few concerns raised by the reviewers have been positively addressed.